# Efficacy of Wholistic Turmeric Supplement on Adenomatous Polyps in Patients with Familial Adenomatous Polyposis—A Randomized, Double-Blinded, Placebo-Controlled Study

**DOI:** 10.3390/genes13122182

**Published:** 2022-11-22

**Authors:** Ophir Gilad, Guy Rosner, Dana Ivancovsky-Wajcman, Reut Zur, Rina Rosin-Arbesfeld, Nathan Gluck, Hana Strul, Dana Lehavi, Vivien Rolfe, Revital Kariv

**Affiliations:** 1Tel Aviv Medical Center, Department of Gastroenterology, Tel Aviv 6423906, Israel; 2Sackler Faculty of Medicine, Tel Aviv University, Tel Aviv 6997801, Israel; 3Pukka Herbs Ltd., Keynsham BS31 2GN, UK

**Keywords:** familial adenomatous polyposis, curcumin, turmeric, chemoprevention

## Abstract

Several studies have demonstrated that curcumin can cause the regression of polyps in familial adenomatous polyposis (FAP), while others have shown negative results. Wholistic turmeric (WT) containing curcumin and additional bioactive compounds may contribute to this effect. We performed a double-blinded, randomized, controlled trial to assess the efficacy of WT in FAP patients. Ten FAP patients were randomly assigned to receive either WT or placebo for 6 months. Colonoscopies were performed at baseline and after 6 months. The polyp number and size, as well as the cumulative polyp burden, were assessed. No differences were noted between the groups in terms of changes from the baseline’s polyp number, size, or burden. However, stratifying the data according to the right vs. left colon indicated a decrease in the median polyp number (from 5.5 to 1.5, *p* = 0.06) and polyp burden (from 24.25 mm to 11.5 mm, *p* = 0.028) in the left colon of the patients in the WT group. The adjusted left polyp number and burden in the WT arm were lower by 5.39 (*p* = 0.034) and 14.68 mm (*p* = 0.059), respectively. Whether WT can be used to reduce the polyp burden of patients with predominantly left-sided polyps remains to be seen; thus, further larger prospective trials are required.

## 1. Introduction

Colorectal cancer (CRC) is a common lethal malignancy, ranked as the third most commonly diagnosed neoplasm in both males and females, and the second leading cause of cancer death worldwide [1,2]. Polyposis syndromes are rare, devastating inherited conditions caused by mutations in cancer-related genes with a phenotype of numerous polyps along the gastrointestinal tract, mainly the colon [3]. Familial adenomatous polyposis (FAP) is an autosomal dominant syndrome, caused by mutation in the adenomatous polyposis coli (*APC*) gene, leading to a lack of degradation of β-catenin in the cytoplasm, which then enters the nucleus and enhances the transcription of oncogenes [4].

FAP patients can present with a classic early onset phenotype that is manifested by hundreds to thousands of adenomas in the colon and the rectum by the second decade of life or by an attenuated, milder phenotype in which polyps develop later in life, with a wide polyp burden range [3,5].

Nearly 100% of FAP patients will develop CRC during their lifetime, if untreated; therefore, patients are required to undergo annual colonoscopies to monitor polyp burden and resect as required. When the polyp burden cannot be managed endoscopically, or if high-grade dysplasia or CRC develops, or patients become symptomatic, or surveillance becomes impossible, colorectal surgery should be performed [3,6]. Surgical options for FAP patients include colectomy with ileorectal anastomosis (IRA), mainly for those patients with low rectal polyp burden, total proctocolectomy with ileal pouch–anal anastomosis (TPC-IPAA) or total proctocolectomy with ileostomy formation [6,7]. Besides the inherent surgical risks found in any major abdominal surgery, these procedures are also associated with morbidity due to increased bowel movements and incontinence [3,8]. While surgery greatly reduces the risk of CRC, patients still require the surveillance of either their rectal stump or ileal pouch [9].

Due to its high penetrance and the ability to monitor adenoma development via endoscopy, FAP has long been studied as a prototype model for CRC development and as an illness suited for chemoprevention interventions [10,11], as successful chemoprevention may enable the delay of surgery, thus shortening the periods with surgery-related complications and delaying post-surgical adenoma progression, so the risks of malignancy and further operations are decreased.

Several chemopreventive agents have been studied in FAP patients, such as aspirin [12], the non-steroidal anti-inflammatory drug (NDSAID) sulindac [13,14,15], and COX-2 inhibitors [16,17]. However, these drugs either failed to show a reduction in the polyp burden, had conflicting results, or are associated with severe adverse events.

Curcumin is one of many constituents of the spice turmeric (*Curcuma longa* L.), alongside flavonoids, turmerones, curlone, and zingiberene. Turmeric is widely used in the practice of complementary medicine in China and India for a variety of ailments [18]. In recent years, there has been increasing evidence regarding the specific value of curcumin as an anti-inflammatory supplement in conditions such as inflammatory bowel disease [19], and as an antineoplastic agent [20,21].

Specifically, much evidence, both in vivo and in vitro, has been gathered regarding curcumin’s ability to prevent colorectal cancer. Possible mechanisms for CRC prevention have been suggested including microbiome modulation, cell cycle arrest, the stimulation of apoptosis, and epigenetic effects [22,23].

In FAP mouse models, curcumin has been shown to reduce polyp number and size [24,25]; however, human studies have shown conflicting results [26,27]. To complicate things further, multiple formulations exist for curcumin, including different delivery systems and combinations with different compounds, so studies with different formulations cannot be easily compared [28]. Besides curcumin, turmeric’s essential oil also contains other bioactive compounds known as turmerones, which may also play a role in cancer prevention [29].

To further investigate the effect wholistic turmeric (WT) has on the polyp burden in FAP patients, we conducted a double-blind, placebo-controlled, randomized trial using WT as the curcumin study formulation.

## 2. Patients and Methods

FAP patients aged 18–70 years at a national tertiary referral center at Tel Aviv Medical Center, with an established clinical diagnosis of familial polyposis, who carry a pathogenic mutation in the *APC* gene were included. The patients were either before or after colonic surgery, mainly subtotal colectomy with IRA or TPC+ IPAA.

The exclusion criteria included 1. pregnant or nursing women; 2. the use of any COX inhibitors for more than 1 week over the prior 3 months; 3. concomitant severe or uncontrolled cardiovascular, hepatic, renal, or metabolic disease; 4. known allergy to curcumin; 5. anticipated surgery within 6 months.

The patients were removed from the study in case of the occurrence of any severe adverse event (SAE) subjected to the primary investigator’s decision, showed intolerance to the study drug, had liver enzyme elevation of more than twice the upper normal limit after 4–8 weeks of treatment, demonstrated significantly abnormal blood count after 4–8 weeks of treatment, or became pregnant.

### 2.1. Study Protocol

The patients were randomly assigned in a 1:1 ratio to either treatment with wholistic turmeric capsules or a placebo.

Each participant received 8 capsules per day (4 capsule bid) for six months. Each oral turmeric capsule contained 123.65 mg curcuminoids and 26.79 mg turmerones. Each capsule also contained pepper fruit, spirulina, seaweed, and ginger root. The patients and investigators were blinded to the randomized intervention. To the best of our knowledge, this is the first study using this formulation. The wholistic turmeric and identical placebo capsules were kindly provided by Pukka Herbs (Keynsham, UK).

The patients were contacted at week 1 and then every 4 weeks for the overall study duration for adverse events, and 1 month and 6 months post-treatment. The blood samples were sent for a complete blood count and chemistry at study initiation, after 4–8 weeks, and at the end of the 6-month period, and the abnormal values were reported.

A colonoscopy was performed at study initiation wherein the polyps were counted and measured with open forceps (7 mm diameter) placed near the polyp to determine its size (see Figure 1). Polyp location was documented as either in the left colon (the rectum up to and including the splenic flexure, or patients with an ileal pouch) or the right colon (at the transverse colon up to the cecum). The polyp burden was calculated as the total sum (mm) of all the polyps in the colon.

A second colonoscopy was performed at study termination (6 months). As in the index colonoscopy, polyp number, size, and colonic location were documented as mentioned above for the baseline colonoscopy. The changes in polyp characteristics (number, average size, maximal size, and total burden) were calculated as the percent of change from the baseline measurements. One physician (R.K.) performed all the endoscopies. In addition, several of the study patients were under surveillance in our department, and data were collected from their endoscopic procedure one year prior to study enrollment. The annual growth rate was calculated pre-intervention (as the percent of change in the baseline characteristics from those one year prior to the intervention) and post-intervention (as the percent of change in the 6-month follow-up characteristics from those at the baseline). The study was ethically approved by the institutional review board. The trial is registered at clinicaltrials.gov (NCT03061591).

### 2.2. Statistical Analysis

The continuous variables (i.e., age, polyp number, size, and burden) are presented as median + interquartile range (IQR) and the dichotomous variables (i.e., gender and operation status) as proportions. The continuous variables were compared between the two groups using the Mann–Whitney test, and the categorical variables were compared between them using a chi-square test or Fisher’s exact test. The baseline’s polyp characteristics (using either polyp number, size, or burden as covariates) were adjusted using the analysis of covariance (ANCOVA). Changes in the variables before and after therapy were assessed using a Wilcoxon signed-rank test for the continuous variables. Spearman’s rho was used to evaluate the correlation of changes between the right- and left-sided polyp number and burden. For all the analyses, *p* < 0.05 was considered statistically significant, and SPSS software was used (IBM SPSSS statistics, IBM Corp., version 25, Armonk, NY, USA, 2017).

## 3. Results

Between 2018 and 2020, eleven patients were recruited for this study. One patient was later excluded from the study due to participation in a different clinical trial for hemophilia. The baseline characteristics of the patients are shown in Table 1. There were no significant differences in demographics or the baseline characteristics of polyps at the initial endoscopy between the two groups. Three patients were post-TPC-IPAA, and seven patients had intact colons. Genetically, all the patients had an identified mutation in the *APC* gene, four with a deletion and six with a missense mutation.

The polyp data for each individual patient before and after intervention (at baseline and 6 months), including polyp number, burden, mean, and maximal size, is shown in Table 2 and Figure 2.

Wide-range heterogeneity in response was observed among the patients, as only 50% of the patients in the WT group demonstrated decreased polyp number and decreased maximal polyp size, and overall, 66.67% had a decrease in the mean polyp size and polyp burden, compared with a similar 50% decrease in polyp number, mean, and maximal polyp size and a 75% decrease in the polyp burden in the placebo group. Changes in polyp number, size, and burden were not significant in either group. In Table 3, the changes in polyp number, burden, mean, and maximal size between the two groups are compared. No significant differences were noted, as both groups experienced a decrease in the median polyp number, burden, mean, and maximal size. The large interquartile range demonstrates the large heterogeneity in responses. Adjusting for the baseline’s polyp number, burden, mean, and maximal size using the analysis of covariance did not alter the results of the study (*p* = 0.342, 0.527, 0.689, and 0.639, respectively), nor did we note any effects of age or gender on outcomes.

After stratifying the data according to polyp response in the right vs. left colon, we did not notice any significant difference between the two groups in all the parameters (Table 4). The distribution and characteristics of the polyps according to the location in each individual patient are shown in Table 5. Using Spearman’s rho to determine whether there was a correlation between the changes in the left and right polyp number and burden produced negative results (r = 0.08, *p* = 0.872 and r = 0.029, *p* = 0.957, respectively). However, we did observe an overall reduction in the polyp size and burden in the left colon in the overall cohort (median 6.5 to 5.5 polyps and 24.5 mm to 19 mm, *p* = 0.028 and 0.017, respectively), which was not apparent in the right colon. When we analyzed the WT group and the placebo group separately, we noticed the same trend only in the WT group, as the median polyp number dropped from 5.5 to 1.5 (*p* = 0.06) and the median polyp burden dropped from 24.25 mm to 11.5 mm (*p* = 0.028) in the left colon. This difference between the baseline and follow-up endoscopies was not significant in the placebo group, nor was it seen in the right colon or for the polyp size in either group.

Taking all of this into account, we examined whether WT affects the left-sided polyp size and burden in the entire cohort after adjusting for the baseline parameters and noted that the adjusted mean polyp number was lower by 5.39 (*p* = 0.034), and the adjusted mean polyp burden was lower by 14.68 mm (*p* = 0.059) in the WT group, compared with the placebo group. Figure 3 illustrates the difference in the change in the left- and right-sided polyp number and burden between the two groups.

There was one patient in the placebo group who had a robust decrease in polyp number (patient 8 in Table 2) from 241 to 108 polyps. Since this is an extreme decrease in polyp number, we investigated whether removing this outlier from our analysis affects our results, but we noted no difference in the trends or statistical significance of our outcomes.

Eight patients were followed in our clinic and underwent a colonoscopy in the year prior to study initiation (five in the WT groups and three in the placebo group). We were able to extract the data regarding polyp number and maximal size from these early endoscopies, calculate the annual growth rate of polyp characteristics, and compare it to the growth rate after study initiation. Since not every single polyp was measured in the endoscopies that were performed prior to the study, there was no way to compare the change in the annual polyp burden and the mean polyp size before and after treatment. Figure 4 illustrates these differences in the growth rates between the WT group and the placebo.

The WT group had an annual decrease of 15% in polyp number before treatment initiation, which dropped to only a 10% decrease after treatment, while the placebo group had a 30% increase in polyp number before the study compared with a 12.8% decrease after treatment; these changes in polyp number were not significantly different between the groups (Wilcoxon signed-rank test, *p* = 0.612). Changes in the maximal polyp size were also not significantly different, as the WT group had an annual increase in the size of 200% before treatment and a 6.25% decrease after treatment, while the placebo group had an increase of 60% before the initiation of therapy compared with a 15% decrease after therapy (*p* = 0.327).

The only adverse events documented were the two patients in the WT who reported minor diarrhea, and no changes in blood work, including their blood count and metabolic panel, were noted.

## 4. Discussion

The optimal chemopreventive agent for CRC is one that can effectively prevent or at least postpone the occurrence of malignancy, has a favorable safety profile with minimal adverse events, is easy to administer, and has a wide availability and low cost. The search for a chemopreventive drug for FAP patients has produced several candidates, though none have answered all the above prerequisites. The reports of the CAPP1 study using aspirin had disappointing results and failed to show a reduction in polyp number or size [12]. Studies using the non-steroidal anti-inflammatory drug sulindac showed conflicting results; while some showed a reduction in the polyp burden, these studies suffer from small cohorts and/or short follow-ups [13,14] and are in conflict with other studies that failed to reproduce these results [15]. In contrast, the selective COX-2 inhibitor celecoxib has been shown to reduce the polyp burden in several studies [16,17]; however, the COX-2 inhibitor family is associated with the increased risk of cardiovascular events, which is a major consideration when prescribing these drugs as preventive tools [30,31]. The mammalian target of rapamycin (mTOR) inhibitors has been shown to reduce the polyp burden in mouse models [32,33], but limited data from humans are impeded by significant toxicities [34], as is the case with the use of sulindac plus erlotinib, an epidermal growth factor receptor antagonist [35].

The use of natural products is, therefore, an appealing strategy to combat cancer development, as these compounds often carry a minimal-side-effect profile, are easily administered, and have a high compliance rate. As a popular herbal supplement in the East, turmeric had been used for centuries for various ailments and diseases, as well as a cancer remedy in the Indian natural medical literature [18]. In recent years, there has been a growing body of knowledge of the beneficial effects the curcumin constituent might have on multiple neoplasms, including prostate, colorectal, breast, brain, and head and neck cancers [36]. For CRC, curcumin has been shown to interact with multiple molecular targets, resulting in the modulation of several signaling pathways [22]. Curcumin has been shown to cause cell cycle arrest by downregulating cyclin D1 [37], inducing reactive oxygen species [38], reducing expression of COX-2 [39], decreasing methylation [40], reducing β-catenin signaling [41], and decreasing the expression of the antiapoptotic protein Bcl-2 [42]. Additionally, curcumin use has been shown to have a significant impact on the microbiome by decreasing the abundance of cancer-related species and increasing that of beneficial bacteria [43].

In-vivo studies of FAP mouse models using curcumin have shown its effectiveness in reducing the polyp burden. One such study showed a dose-dependent effect of curcumin, as curcumin at 0.1% in the diet was without effect, while at 0.2% and 0.5%, it reduced adenoma growth by 39–40% [24]. Another mouse model using a combination of plant-based substances including curcumin, silymarin, and boswellic acid noted a decrease in polyp number and size, the areas of low-grade dysplasia, and intestinal carcinoma [25].

Human studies, however, have shown conflicting results. An early study treated 41 smokers with 98% pure curcumin 2 to 4 g per day for 30 days and evaluated the number of aberrant crypt foci. The study noted no effect of the 2 g dosage, while a significant reduction in aberrant crypt foci was noted with the 4 g dosage [44]. The first study on FAP patients by Cruz-Correa et al. treated five post-colectomy patients with 1440 mg of curcumin and 60 mg of quercetin, another plant-derived antioxidant, daily for 6 months. A decrease in polyp number was noted in 4/5 patients after 3 months and 4/4 patients at 6 months (one patient lost to follow-up), with an overall decrease in polyp number, by 60.4%, and a decrease in polyp size by 50.9% [26]. However, a more recent, larger randomized, controlled trial by the same authors had a negative outcome [27]. The authors randomized 44 FAP patients with either intact colon or post-surgery (IRA or IPAA) to either pure curcumin 3 g per day or a placebo for 12 months. At the end of the study, there were no significant differences in the mean number of polyps, polyp size, polyp burden, or change in the number of polyps from the baseline. It is worth mentioning that the two studies by Cruz-Correa used slightly different formulations, as the first pilot study used almost half the dosage of curcumin of the second study but in a formulation that contained piperine, a component of black pepper, and in combination with the antioxidant quercetin [27].

In the current study, we used wholistic turmeric as the study drug. This formulation, while containing a lower concentration of curcuminoids, uses an extraction technique (“wholistic”) to retain a wider range of active constituents such as flavonoids and essential oils; this includes other phytochemicals thought to have protective effects against colorectal cancer such as α-turmerones. In a model using induced carcinogenesis on the background of mouse colitis, the use of turmerones decreased COX-2 expression and reduced adenoma growth by 73%, while the combination of both curcumin and turmerone abolished colon tumor formation altogether [29]. We, therefore, hypothesized that the synergism of the various compounds found in WT would improve the polyp characteristics of FAP patients. However, our study’s results are in concordance with Cruz-Correa’s later study, as we noted no significant differences in polyp number, size, or burden after 6 months of treatment.

There are several potential explanations for our results. First, as mentioned above, curcumin has a dose-dependent response [24], so a low dosage of the drug might explain the lack of response. The patients in our study received a little less than 1 g of curcumin per day, which is lower than the dosage used in other studies, as we expected a potential synergistic effect of the various compounds found in WT. Secondly, curcumin has numerous derivatives with varying antineoplastic effects, with some modified synthetic curcuminoids shown to enhance anticancer activity compared with others [36,45,46]. The formulation that was used in our study used wholistic turmeric with no synthetic curcuminoids which might themselves produce more favorable outcomes than natural curcuminoids. Third, there are several curcumin delivery systems available, including polymeric nanoparticles, nanogels, liposomes, or peptide/protein complexes. These different formulations have different levels of bioavailability and thus different antineoplastic effects [36]. The WT used in our study did not use any specialized delivery system, which might reduce its function. Fourth, the benefits of curcumin were often noted when used in conjunction with other phytochemicals, for example, the FAP mouse model noted above that utilized curcumin together with the phytoestrogen silymarin and boswellic acid, both of which have been previously shown to hamper carcinoma development in mouse models [25]. Cruz-Correa’s first study, which showed a positive effect of curcumin, also used the flavonoid quercetin known for its antioxidant properties [26]. The author later noted that the formulation in this pilot study also contained piperine, a component of black pepper and a known inhibitor of hepatic glucuronidation that increases the bioavailability of curcumin by 2000% [47]. While the formulation used in our study did not contain silymarin, boswellic acid, or quercetin, it did contain 31 mg of pepper fruit extract, but this dosage might not have supplied our patients with enough piperine. Furthermore, the WT formulation used in our study contained turmerones, which have demonstrated their effectiveness in preventing CRC in mouse models, but these models used CRC, which developed from a background of colitis [29] rather than an inherited mutation in the tumor suppressor gene *APC*.

Although our study failed to show a clinical benefit of WT in the overall cohort, we did recognize a noticeable decrease in the polyp size and burden in the left colon. Several studies have previously demonstrated that several molecular pathways are expressed differently in the left and right CRC; for example, the components of the phosphatidylinositol 3-kinase/Akt/mammalian target of the rapamycin signaling axis are overexpressed in the left-sided CRC [48], mutations in *TP53* are 1.5–3 fold more frequent in distal tumors, and microsatellite instability (MSI) phenotype is up to 10 times more frequent in proximal tumors [49]. It has also been previously shown that several agents act differently upon the different parts of the colon; for example, Ishikawa et al. demonstrated that low-dose aspirin for chemoprevention significantly reduced the diameter of the polyps removed from the descending and sigmoid colon of FAP patients [50], and conventional chemotherapies and targeted therapies are more effective in left-sided CRC, while MSI-high tumors, which are more common in the right colon, benefit more from immunotherapies [51]. It is, therefore, possible that WT affects the molecular pathways that are more commonly expressed in left-sided polyps, rather than the entire colon. Our findings support the potential use of curcumin in patients with FAP after subtotal colectomy and IRA as a potential subgroup for future studies.

Another interesting phenomenon we noticed in our study was a spontaneous regression of polyps in some of our patients, as we noted that half of our placebo patients had a decrease in polyp number during our investigation, and the WT group had a 15% decrease in polyp number in the year prior to intervention. The theory that some polyps regress over time is not new and has been hypothesized before as the reason why we do not see a higher incidence of CRC, while the adenoma detection rate in endoscopies increases [52], and have also been demonstrated before in an older study by Knoernschild who tattooed the mucosa near asymptomatic benign polyps and observed that in 18% of patients, the polyps completely disappeared after 3–5 years [53].

Our study has some limitations. FAP is rare, and single-center studies include typically small cohorts. The cohort size of 10 patients is underpowering the study and might have different results had more patients been recruited. A small cohort can also cause outliers to have a more significant effect on results, although removing these outliers did not eventually alter the outcomes of our study. The small sample size also makes it difficult to assess whether certain subgroups in the study might have benefitted from curcumin more than others. The short follow-up makes it difficult to draw long-term conclusions. Our study also did not take into account other factors that may affect the occurrence of polyps such as diet, smoking, or obesity.

In conclusion, our study did not demonstrate any beneficial effects of wholistic turmeric on polyp number, size, or burden in FAP patients within a treatment period of 6 months. On the other hand, we showed a significant and notable effect on the left-sided polyps with a reduction in their number and burden. Whether WT can be used to reduce the polyp burden of patients with predominantly left-sided polyps or be effective in patients after subtotal colectomy remains to be seen. Different formulations with different delivery systems of curcumin might have different results, but larger prospective trials are required to further study the effect of wholistic turmeric on colorectal neoplasia.

## Figures and Tables

**Figure 1 genes-13-02182-f001:**
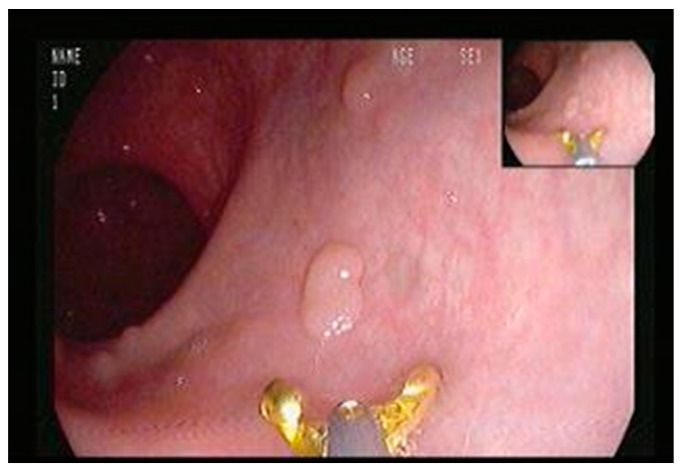
Polyp measurement using open forceps.

**Figure 2 genes-13-02182-f002:**
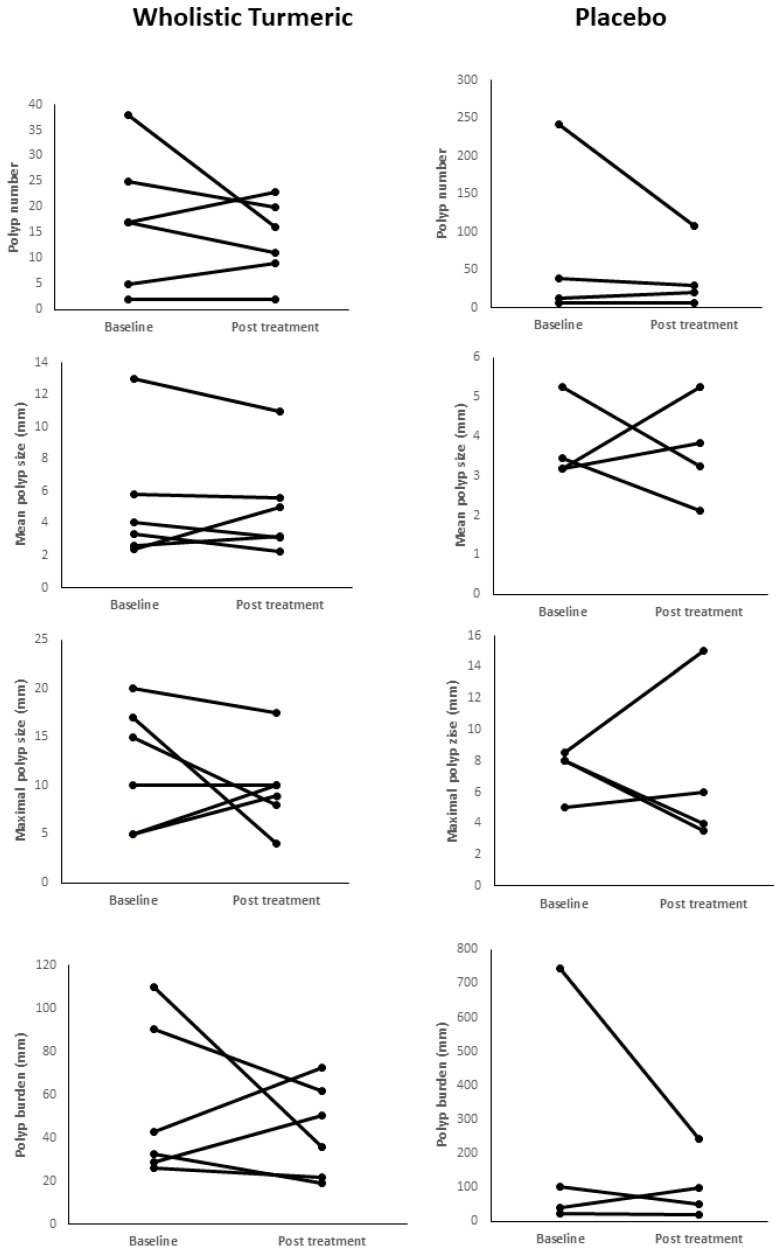
Individual polyp characteristics at baseline and post-treatment.

**Figure 3 genes-13-02182-f003:**
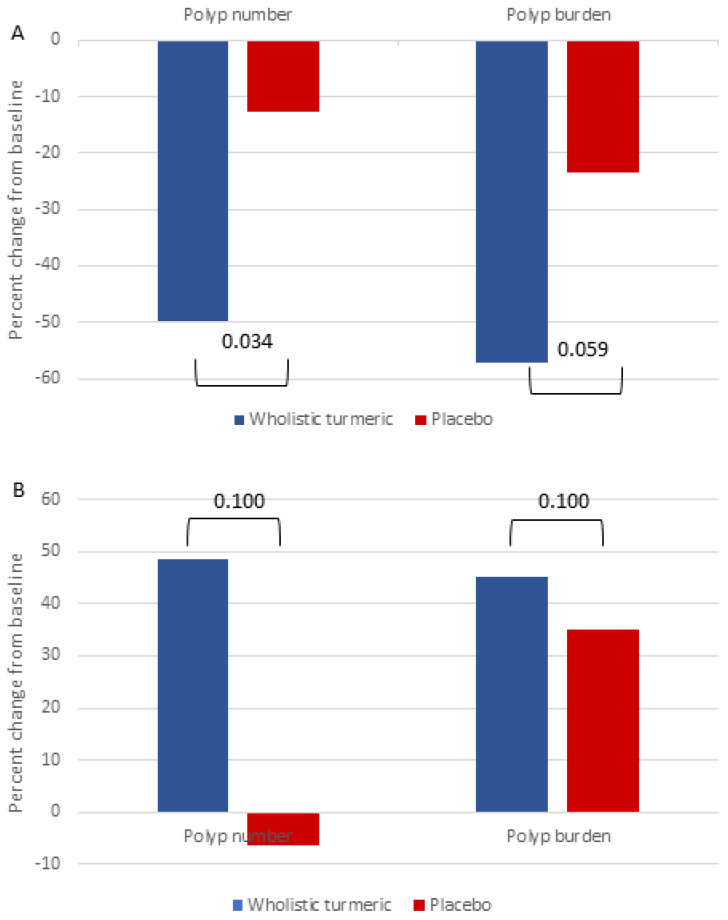
Changes in polyp number and burden (%) in (**A**) the left and (**B**) the right colon for the wholistic turmeric and placebo groups.

**Figure 4 genes-13-02182-f004:**
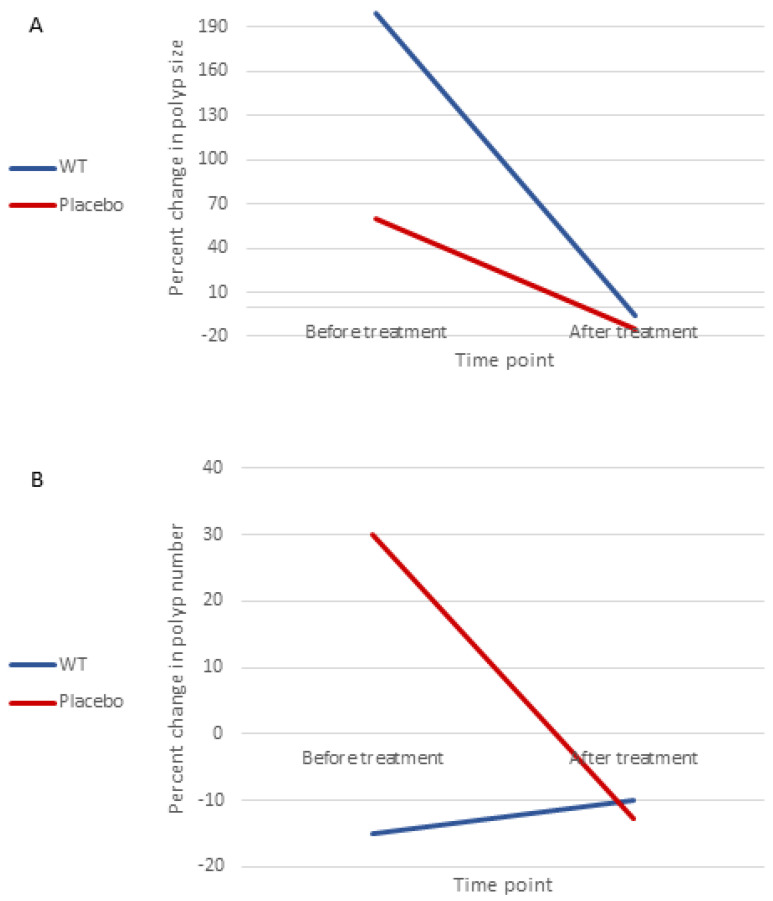
Change in the growth rate of (**A**) maximal polyp size and (**B**) polyp number by treatment groups. WT, wholistic turmeric.

**Table 1 genes-13-02182-t001:** Baseline characteristics of patients randomized in the study (median + IQR unless otherwise stated).

	Wholistic Turmeric N = 6	Placebo N = 4	*p* Value
Age *	45 (27–62.25)	45 (41–48)	0.944
Gender (%male) **	66.67	75.00	0.635
Underwent IPAA (%) **	1 (16.67)	2 (50)	0.500
**First endoscopy**			
Number of polyps *	17 (8–23)	25.5 (10.75–89.5)	0.476
Polyp burden (mm) *	37.75 (29.87–78.62)	69.75 (34.62–261.75)	0.762
Mean polyp size (mm) *	3.70 (2.79–5.36)	3.32 (3.2–3.9)	0.721
Maximal polyp size (mm) *	12.5 (6.25–16.5)	8 (7.25–8.12)	0.352

IPAA, ileal pouch–anal anastomosis. * Mann–Whitney test; ** Fisher’s exact test.

**Table 2 genes-13-02182-t002:** Individual patients’ outcome of polyp number, burden, and maximal size at baseline and follow-up (6 months).

Patient Number	Treatment Group	Underwent Surgery	Polyp Number			Mean Polyp Size (mm)			Maximal Polyp Size (mm)			Polyp Burden (mm)		
			Baseline	Follow-Up	Change (%) from Baseline	Baseline	Follow-Up	Change (%) from Baseline	Baseline	Follow-Up	Change (%) from Baseline	Baseline	Follow-Up	Change (%) from Baseline
1	WT	IPAA	2	2	0.0	13	11	**−15.3**	20	17.5	**−12.5**	26	22	**−15.4**
2	WT	No	24	20	**−16.7**	4.07	3.1	**−23.9**	15	8	**−46.7**	90.5	62	**−31.5**
3	WT	No	5	9	80.0	5.8	5.6	**−3.4**	10	10	0	29	50.5	74.1
4	WT	No	17	11	**−35.3**	2.3	5	111.8	5	9	80.0	32.5	19	**−41.5**
5	WT	No	17	23	35.3	2.6	3.1	21.8	5	10	100	43	72.5	68.6
6	WT	No	38	16	**−57.9**	3.3	2.2	**−33.5**	17	4	**−76.5**	110	36	**−67.3**
7	Placebo	IPAA	7	7	0.0	3.4	2.1	**−38.4**	8	4	**−50.0**	23	19	**−17.4**
8	Placebo	No	241	108	**−55.2**	3.2	5.2	64.06	8	3.5	**−56.3**	744.5	244.5	**−67.2**
9	Placebo	No	12	20	66.7	3.2	3.8	19.68	8.5	15	76.5	38.5	98.5	155.8
10	Placebo	IPAA	39	29	**−25.6**	5.2	3.2	**−38.09**	5	6	20	101	50.5	**−50.0**

IPAA, ileal pouch–anal anastomosis; WT, wholistic turmeric.

**Table 3 genes-13-02182-t003:** Comparison of changes in polyp characteristics between placebo and wholistic turmeric.

Median Change (%) from Baseline (IQR)	Wholistic Turmeric N = 6	Placebo N = 4	*p* Value
**Polyp number**	−10 (−31.47–26.47)	−12.8 (−33.02–16.67)	0.896
**Mean polyp size (mm)**	−9.41 (−21.79–15.51)	−9.20 (−38.17–30.78)	0.610
**Maximal polyp size (mm)**	−6.25 (−38.125–60)	−15 (−51.56–34.11)	0.826
**Polyp burden (mm)**	−23.43 (−39.02–47.6)	−33.69 (−54.3–25.91)	0.886

Mann–Whitney test.

**Table 4 genes-13-02182-t004:** Comparison of changes in polyp characteristics between placebo and wholistic turmeric by polyp location.

Median Change (%) from Baseline (IQR)	Wholistic Turmeric N = 6	Placebo N = 4	*p* Value
**Left colon**			
**Polyp number**	−49.78 (−78.57–[−8.82])	−12.82 (−30.1–75)	0.257
**Mean polyp size (mm)**	−34.61 (−43.97–[−19.87])	−2.74 (−27.7–20.31)	0.476
**Polyp burden (mm)**	−57.13 (−86.51–[−35.38])	−23.51 (−34.72–86.95)	0.171
**Right colon**			
**Polyp number**	48.61 (−5.83–93.75)	−6.24 (−30.09–19.6)	0.533
**Mean polyp size (mm)**	−8.07 (−19–5.31)	−64.45 (−73.58–[−55.31])	0.133
**Polyp burden (mm)**	45.2 (−20.18–111.05)	34.92 (−18.84–88.69)	1

Mann–Whitney test.

**Table 5 genes-13-02182-t005:** Individual patients’ outcomes of polyp number and burden divided into right and left colon.

			Right Colon						Left Colon					
			Polyp Number			Polyp Burden (mm)			Polyp Number			Polyp Burden (mm)		
Patient Number	Treatment Group	Underwent Surgery	Baseline	Follow-Up	Change (%) from Baseline	Baseline	Follow-Up	Change (%) from Baseline	Baseline	Follow-Up	Change (%) from Baseline	Baseline	Follow-Up	Change (%) from Baseline
1	WT	IPAA							2	2	0.0	26	22	**−15.4**
2	WT	No	18	19	5.6	68	60	**−11.8**	6	1	**−83.3**	22.5	2	**−91.1**
3	WT	No	4	7	75	23	46.5	102.2	1	2	100	6	4	**−33.3**
4	WT	No	0	0	0	0	0	0	17	11	**−35.3**	32.5	19	**−41.5**
5	WT	No	12	23	91.7	30.5	72.5	137.7	5	0	**−100**	12.5	0	**−100.0**
6	WT	No	10	6	**−40.0**	22	12	**−45.5**	28	10	**−64.3**	88	24	**−72.7**
7	Placebo	IPAA							7	7	0.0	23	19	**−17.4**
8	Placebo	No	195	82	**−57.9**	650	178	**−72.2**	46	26	**−43.5**	94.5	66.5	**−29.3**
9	Placebo	No	11	16	45.5	36.5	88.5	142.5	1	4	300.0	2	10	400.0
10	Placebo	IPAA							39	29	**−25.6**	101	50.5	**−50.0**

IPAA, ileal pouch–anal anastomosis; WT, wholistic turmeric.

## Data Availability

The data presented in this study are available on request from the corresponding author.

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
