# Peer review of "Efficacy of Wholistic Turmeric Supplement on Adenomatous Polyps in Patients with Familial Adenomatous Polyposis—A Randomized, Double-Blinded, Placebo-Controlled Study"

_genes, 2022, doi:10.3390/genes13122182_

Round 1

Reviewer 1 Report

  • FAP and cancer colon is really important topic now days, You paper toke us to a different area of thinking, Not only the traditional way in management, And this is how research should be, In the last few years there was a lot of argument about the use of Turmeric and Im happy to review you paper.
  • General concept comments 
    Article: The main area of weakness I this study is the small number, Which affects the power of the study and the accuracy of the final outcome, Also some details were missing from the methodology 
    Review: Your paper will be more clear if you you add some details about the way of counting the polyps , Explain why you chose this specific formulation and dose. And why 6 months of follow up was your target.
  • Specific comments
  •  Line 88: need to mention more about the risk factors that may affects your outcome, Like smoking, Quality of life, Diet 
  • 96: explain why this dose and why this formula 
  • 110-111: clarify the way used to count the polyps and add photos if possible 
  • 186: This is really important to mention at the discussion section 
  • Only 20% of the references were new >2019, Please add more new references 

Reviewer 2 Report

Manuscript title "Efficacy of Wholistic Turmeric supplement on adenomatous polyps in patients with Familial Adenomatous Polyposis- A randomized, double blinded, placebo-controlled study" focuses on use of turmeric as supplement for betterment of FAP. Although the paper appeared to be well-written and presented, several revisions were required before it could be processed further.
Suggestions for improvement on paper
Minor
1. Include a discussion of the potential mechanisms by which turmeric may exert an effect
on adenomatous polyps.
2. Discuss the implications of the findings in light of the current understanding of the etiology
and pathogenesis of familial adenomatous polyposis.
3. Discuss the potential clinical implications of the findings, including the possibility of using
turmeric supplements as a preventative measure or treatment for familial adenomatous
polyposis.
4. Include a discussion of the limitations of the study, including the small sample size and the
lack of long-term follow-up.
5. Future directions for research on the efficacy of turmeric supplements for the treatment of
familial adenomatous polyposis should be discussed.
Major
1. As the Curcumin is main factor behind all the activities that turmeric posses, what
dose of Curcumin is there in capsule of turmeric ?
2. Why that much dose is required and how did you choose the dose.
3. It surely capable of reducing the size of FAP, but what about their effect on liver
functions, as high doses of Curcumin is needed to digest.
4. Did you perform metabolic profiling of the formulation.
5. Is there any chances of recurrence after suppliment stopped.
6. How much dose was bioavailable ??
7. Did you studied the Pharmacokinetics of the administered formulation.

Reviewer 3 Report

 Efficacy of Wholistic Turmeric supplement on adenomatous polyps in patients with Familial Adenomatous Polyposis- A randomized, double blinded, placebo-controlled study

This is an interesting study which talks about the use of curcumin formulation in familial adenomatous polyposis (FAP) treatment. Although the study could not result in clear positive outcomes, an elaborate and critical discussion of results is provided. The discussion section makes strength of this manuscript. Following are the few suggestions to improve the quality/readability of manuscript.

Major Comments

1.      On Line 95 it says Patients were randomly assigned in a 1:1 ratio to either treatment with wholistic turmeric capsules or placebo. Please include the reason for the different N for WT and control group (6 and 4 respectively) on all the Tables.

 2.      As the title says, this study is about efficacy of Wholistic Turmeric as a supplement, were there any concurrent medications taken by patients during study duration. Please specify.

 3.      In addition to Table 4 can we get a subject-wise comparison of left and right colon polyp data? It would make it clear if the decrease in polyp burden in left colon had any association with increase in polyp burden in right colon for the same patient.

 4.      If the study duration was 6 months or one year? Please elaborate why the study protocol says study termination at 6 months whereas Figure 3 shows annual data. 

 5.      On Line 118 it says, “several patients were under surveillance in our department and data was collected from their endoscopic procedure one year prior to study enrollment”. Is this the pool of patients out of which study group (of 10 patients) was sampled out based on selection criteria? If so, please add this information in the beginning of protocol on Line 95. If not, please mention how that group contributed to the study.

 6.      “Patients were either before or after colonic surgery” on Line 83. Which patients were enrolled before, and which were enrolled after colonic surgery. Please add an additional column on table 2 to indicate patient enrollment was either before or after surgery and if it was IRA or TPC+ IPAA.

 7.      The Age group (18-70 years) mentioned on Line 81 seems to be too broad. Was collected data enough to show any impact of age on treatment outcome?

 8.      Various statistical tests were performed to make conclusions. Adding specific test names in relevant sections and table/figure captions would help readers to better understand the statistical analyses.

 9.      Please check if the correct term for variable type is “categorical” or “categorial”.  Add the names of tested continuous and categorical variables under ‘Statistical analysis’ section. Also, mention what covariates were tested for ANCOVA analysis.

 10.   Is there any other study published using WT or this is the first ever study with this formulation? Please mention in the protocol where composition of WT is discussed.

Minor Comments

1.      Please make a separate paragraph for WT formulation details mentioned on Line 97 to highlight this information. (Probably merge it with sentence on line 100). If it is an approved marketed tablet formulation, a label can be given for the reference.

 2.      As Wholistic Turmeric (WT) contains curcumin and additional bioactive compounds it is more appropriate to call it as a “curcumin study formulation” instead of “study drug” on Line 79.

 3.      Please use either WT or Wholistic Curcumin in place of curcumin throughout the manuscript (e.g., in Table 3 title).

 4.      As discussed on Line 72 that different curcumin formulations had shown different efficacy against polyps, authors may want to make a list to summarize different curcumin formulations/delivery systems, administration routes, animal models and study outcomes (Positive, Negative or No Effect). It might help shed light on whether it is route or delivery system or composition of formulation that produces different treatment outcomes.  

 5.      Please elaborate what does word consecutive mean in “Consecutive FAP patients” on Line 81.

 6.      It should be read as dose-dependent response instead of dose response on Line 280.

 7.      The APC gene should be called out at first appearance in the manuscript.

 8.      What does it mean by “total sum (mm) of all polyps” on Line 110. Is it the sum of size or diameter of all polyps?

 9.      Please correct “1 week and then every 4 weeks” to “week 1 and then every 4 weeks” on Line 102.

 10.   Please correct different Text style on Line 130 and 131.

Round 2

Reviewer 2 Report

Recheck all the statistical information and confirm the interpretations of all the acronyms. I think it appears acceptable to move forward.